# Structural Defects in TiNi-Based Alloys after Warm ECAP

**Aleksandr Lotkov [1],\*** , **Anatoly Baturin [1]**, **Vladimir Kopylov [2]**, **Victor Grishkov [1]** and **Roman Laptev [3]**

[1] Institute of Strength Physics and Materials Science of the Siberian Branch of the Russian Academy of Sciences, 634055 Tomsk, Russia; abat@ispms.tsc.ru (A.B.); grish@ispms.tsc.ru (V.G.)

[2] Physical-Technical Institute of the National Academy of Sciences of Belarus State Scientific Institution, 220141 Minsk, Belarus; kopylov.ecap@gmail.com

[3] National Research Tomsk Polytechnic University, 634050 Tomsk, Russia; laptev.roman@gmail.com

\* Correspondence: lotkov@ispms.ru; Tel.: +7-913-850-0310

**Abstract:** The microstructure, martensitic transformations and crystal structure defects in the $Ti_{50}Ni_{47.3}Fe_{2.7}$ (at%) alloy after equal-channel angular pressing (ECAP, angle 90°, route $B_C$, 1–3 passes at T = 723 K) have been investigated. A homogeneous submicrocrystalline (SMC) structure (grains/subgrains about 300 nm) is observed after 3 ECAP passes. Crystal structure defects in the $Ti_{49.4}Ni_{50.6}$ (at%) alloy (8 ECAP passes, angle 120°, $B_C$ route, T = 723 K, grains/subgrains about 300 nm) and $Ti_{50}Ni_{47.3}Fe_{2.7}$ (at%) alloy with SMC B2 structures after ECAP were studied by positron lifetime spectroscopy at the room temperature. The single component with the positron lifetime $\tau_1 = 132$ ps and $\tau_1 = 140$ ps were observed for positron lifetime spectra (PLS) obtained from ternary and binary, correspondingly, annealed alloys with coarse-grained structures. This $\tau_1$ values correspond to the lifetime of delocalized positrons in defect-free B2 phase. The two component PLS were found for all samples exposed by ECAP. The component with $\tau_2 = 160$ ps (annihilation of positrons trapped by dislocations) is observed for all samples after 1–8 ECAP passes. The component with $\tau_3 = 305$ ps (annihilation of positrons trapped by vacancy nanoclusters) was detected only after the first ECAP pass. The component with $\tau_3 = 200$ ps (annihilation of positrons trapped by vacancies in the Ti sublattice of B2 structure) is observed for all samples after 3–8 ECAP passes.

**Keywords:** TiNi-based alloys; ECAP; microstructure; positron lifetime spectroscopy; nanoclusters; vacancies; dislocations

## 1. Introduction

Severe plastic deformation (SPD) allows one to produce an ultrafine-grained (UFG) structure with a high strength and sufficient ductility in metal materials [1,2]. Among such materials are TiNi-based alloys distinguished for their superelasticity and shape memory effect (SME) and widely used in engineering and medicine [3]. Among the SPD methods is equal channel angular pressing (ECAP), which provides UFG TiNi-based alloys with greatly improved functional properties. It opens the most promising way of producing bulk billets from this type of materials [1,2].

By now, research data are available to judge the microstructure evolution of binary and ternary TiNi-based alloys during ECAP at 623–773 K, and the grain size effect on their mechanical and inelastic properties [4–17], and also the influence of ECAP on the temperatures of martensite transformations (MT) in TiNi-based alloys [9,10,12,17,18]. As has been shown [1,4,12], the grain structure of TiNi-based alloys can be refined to 100–500 nm in six to eight ECAP passes depending on the pressing temperature. Such alloys with an average grain-subgrain size of 300 nm show an unusual behavior with an ultimate



strength of up 1200 MPa and plasticity of up to 50–60%. After ECAP at 623–773 K, the temperature of martensite transformations in TiNi-based alloys normally decreases by 20 K and more [17,18]. However, very scant data are available on the evolution of lattice defects in TiNi-based alloys exposed to ECAP. These data are necessary for the understanding both the mechanisms of grain refinement and physical factors affected on MT temperatures in alloys after ECAP. But the sole paper report that the dislocation density in such alloys after SPD increases greatly (to $\sim 10^{15}$–$10^{16}$ m$^{-2}$) and that its critical value ($\sim 10^{18}$ m$^{-2}$) exists at which they assume an amorphous structure [19].

However, in addition to dislocations, SPD produces numerous excess vacancies as against thermodynamic equilibrium [20,21]. On the one hand, vacancies greatly accelerate the mass transfer, dissolution or precipitation of secondary phases [22,23], and generation of new grain boundaries in a material [24]. On the other hand, vacancies with a high concentration and low mobility can form clusters [25], and eventually decrease the long-term durability of materials, as it happens, e.g., in Al- and Ti-based alloys after ECAP [26,27]. In this context, studying the formation of vacancy-like free volumes in SPD materials can provide a better understanding of the mechanisms responsible for their properties in UFG states and of the conditions for their more efficient use. Most of the studies of SPD-induced vacancies have been performed on pure metals and their alloys [20,28–32], and almost all of them consider SPD processes at room temperature when the vacancy mobility is low. Among the methods of research in deformation-induced vacancies, their clusters, and dislocation density in metals and alloys are resistometry [21], dilatometry [29], differential scanning calorimetry [21,29], X-ray line profile analysis [33], perturbed $\gamma$-$\gamma$ angular correlation (PAC) [34], and positron annihilation spectroscopy (PAS) [28–32], all of which give comparable vacancy concentrations for pure metals ($10^{-2}$–$10^{-4}$). However, little is known about SPD-induced vacancies in intermetallic compounds where their types are more diverse, compared to metals [35]. For example, vacancy-like free volumes classifiable as interface defects are detected by PAS in ball-milled nanocrystalline Fe3Si [36]. In B2 FeAl under high pressure torsion, the vacancy density can reach $10^{-2}$ as it follows from comparative data of differential scanning calorimetry and calculations [37]. From PAC data [38], the types of defects that dominate after ball milling are Schottky pairs in PdIn, triple defects in NiAl and FeRh, and antisite defects in FeAl. One of the studies shows that after ultrasonic shock treatment at room temperature, the relative concentration of single vacancies in equiatomic TiNi surface layers increases to $\sim 10^{-5}$ [39,40].

This paper presents the results of complex research into the ECAP effect at 723 K on the microstructure, martensite transformations, and lattice defect evolution in Ti$_{50}$Ni$_{47.3}$Fe$_{2.7}$ (at%) alloy combined with experimental results about lattice defects in Ti$_{49.4}$Ni$_{50.6}$ alloy (at%) alloy after eight ECAP passes at the same temperature.

The choice of experimental alloys is due to the following considerations based on the results of studies of the microstructure and martensitic transformations. These alloys have the B2 phase structure at room temperature both in the initial coarse-grained state and in the submicrocrystalline state. In addition, the grain-subgrain structures of these alloys are similar after ECAP. This makes it possible to study only those defects of the crystal structure that are formed during ECAP at 723 K and to avoid the influence of defects that may appear as a result of MT.

## 2. Materials and Methods

The test Ti$_{50}$Ni$_{47.3}$Fe$_{2.7}$ (at%) alloy were supplied as rotary forged (1220 K) round bars of diameter 25 mm and length 140 mm (MATEK-SMA Ltd., Moscow, Russia). The round bars were forged to a square of $16 \times 16$ mm$^2$ at $1073 \pm 100$ K with further annealing at 773 K for 3 h. Then, the square bars were cut to $14 \times 14$ mm$^2$ and were exposed to ECAP (B$_C$ route) with a channel angle 90°. The number of cycles was N$_i$, where *i* = 1, 2, 3. The test specimens of Ti$_{50}$Ni$_{47.3}$Fe$_{2.7}$ (at%) were pressed at the Physical-Technical Institute NASB (Minsk, Belarus). The initial coarse-grained alloy had a B2 grain size of 20–40 μm. At temperatures above 275 K, its structure was represented by a single B2 phase (the high temperature cubic phase). The sequence and the temperatures of martensite transformations were studied by temperature resistometry (four-point scheme) in cooling–heating cycles.

The test alloy $Ti_{49.4}Ni_{50.6}$ (at%) was supplied by Intrinsic Devices Inc. (San Francisco, CA, USA). In the alloy quenched from 1073 K, the start temperature of direct B2 → B19′ martensite transformation was $M_S = 288$ K (B19′-the monoclinic martensite phase). At room temperature, its initial coarse-grained state was represented by a B2 structure with an average grain size of 50 μm. Its submicrocrystalline state with an average grain-subgrain size of 300 nm was formed by ECAP at T = 723 K at the Ufa State Aviation Technical University (Ufa, Russia). The channel angle was 110° ($B_C$ route), and the number of passes was $n = 8$ [6].

The samples of dimensions $10 \times 10 \times 1$ mm$^3$ for the positron annihilation spectroscopy and bars of dimensions $1 \times 1 \times 20$ mm$^3$ for the electrical resistivity measurements were prepared from bullets of initial alloy and alloy after ECAP. The foils for the transmission electron microscopy (TEM) were prepared by the mechanical polishing with subsequent electrochemical thinning of plates with initial thickness about 0.3 mm.

The microstructure of the test specimens was examined by transmission electron microscopy (JEM-2100, JEOL Ltd., Tokyo, Japan) in Nanotech Shared Use Center, ISPMS SB RAS).

The crystal defects appearing after ECAP were analyzed by positron annihilation spectroscopy (PAS) based on analysis of the positron lifetime spectra. A time resolution of spectrometer is 240 ps [41]. The positron source was $^{44}$Ti with an activity of 1 MBq placed between two identical specimens. The positron beam diameter was about 6 mm. The total number of annihilation events recorded per each spectrum was no less than $5 \times 10^6$. Each specimen was analyzed from three independent the positron lifetime spectra. Processing of the spectra was carried out using LT software [42]. After subtraction of the component related to annihilation in the positron source and background, the spectra could be reliably decomposed into one lifetime component (initial state) or into two (deformed state),

$$S(t) = (I_1/\tau_1)\exp(-t/\tau_1) + (I_2/\tau_2)\exp(-t/\tau_2), \tag{1}$$

where $\tau_1$, $I_1$ and $\tau_2$, $I_2$ are the respective component lifetimes and intensities to judge the type of a defect and its concentration.

The Vickers microhardness was measured on a Duramin-5 microhardness tester (Struers, Ballerup, Denmark) at room temperature under a load of 100 g.

## 3. Results and Discussion

### 3.1. Effect of Warm ECAP on Martensite Transformation Temperatures in $Ti_{50}Ni_{47.3}Fe_{2.7}$

Figure 1 shows typical temperature dependences of resistivity on heating and cooling for $Ti_{50}Ni_{47.3}Fe_{2.7}$ (at%) alloy before and after ECAP passes. These $\rho(T)$ dependences are qualitatively similar for both initial alloy state and after ECAP.

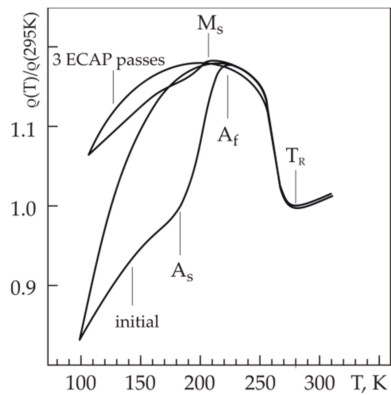

**Figure 1.** Typical temperature dependences of resistivity for $Ti_{50}Ni_{47.3}Fe_{2.7}$ before (initial), and after, three ECAP passes.

On cooling and heating in the temperature range from 275 K to 77 K (liquid nitrogen), the alloy is involved in the sequence of martensite transformations B2 ↔ R ↔ R + B19′ (R and B19′ are rhombohedral and monoclinic martensite phases, respectively). During B2 ↔ R transformations on cooling and heating, the resistivity of samples varies almost without hysteresis (its value is less than 3 K). The transformation R → B19′ on cooling to 77 K is incomplete throughout the volumes of all samples. The MT temperatures are denoted as $T_R$ for B2 ↔ R MT, $M_S$ for the start of R → B19′ MT (cooling) and $A_S$ and $A_f$ for B19′ → R MT (heating) The variations of MT temperatures versus ECAP passes are presented in Table 1.

**Table 1.** Martensite transformation temperatures in $Ti_{50}Ni_{47.3}Fe_{2.7}$.

| State | $T_R$, K | $M_s$, K | $A_s$, K | $A_f$, K |
|---|---|---|---|---|
| Initial | 275 | 213 | 184 | 220 |
| 1st ECAP pass | 276 | 215 | 185 | 208 |
| 3rd ECAP pass | 275 | 195 | 195 | 205 |

The main ECAP effect on MT consists in the following: it decreases $M_S$ by 18 K, compared to the initial state, and narrows the temperature interval of reverse B19′ → R transformation by 10 K. Another effect, which is more pronounced, is a steep decrease in the value by which the resistivity changes during on cooling to 77 K. Therefore, the B19′ volume fraction appearing at T < *Ms* decreases greatly (2–3 times) with increasing the number of ECAP passes, i.e., with increasing the true strain. This is likely due to considerable grain-subgrain refinement in the specimens, which causes their hardening and markedly decreases the martensite transformation temperatures of the R → B19′ MT.

Thus, the B2 structure is observed at room temperature in $Ti_{50}Ni_{47.3}Fe_{2.7}$ (at%) alloy after 1–3 ECAP passes. These data correspond to the results obtained by TEM.

### 3.2. Microstructure of $Ti_{50}Ni_{47.3}Fe_{2.7}$ after Warm ECAP

Figure 2 shows typical micrographs of $Ti_{50}Ni_{47.3}Fe_{2.7}$ (at%) after one ECAP pass. As can be seen, its grain-subgrain microstructure is rather inhomogeneous. In the bulk of the deformed alloy, a clearly defined multiband structure is observed as evidence of plastic strain localization in the specimens during ECAP. The orientation angles of bands relative to each other are 45° (135°) and 90°. Inside the bands, non-equiaxial fragments of a finer grain-subgrain structure are localized. Their aspect ratio (ratio of minimum to maximum sizes) ranges from 1:5 to 1:10.

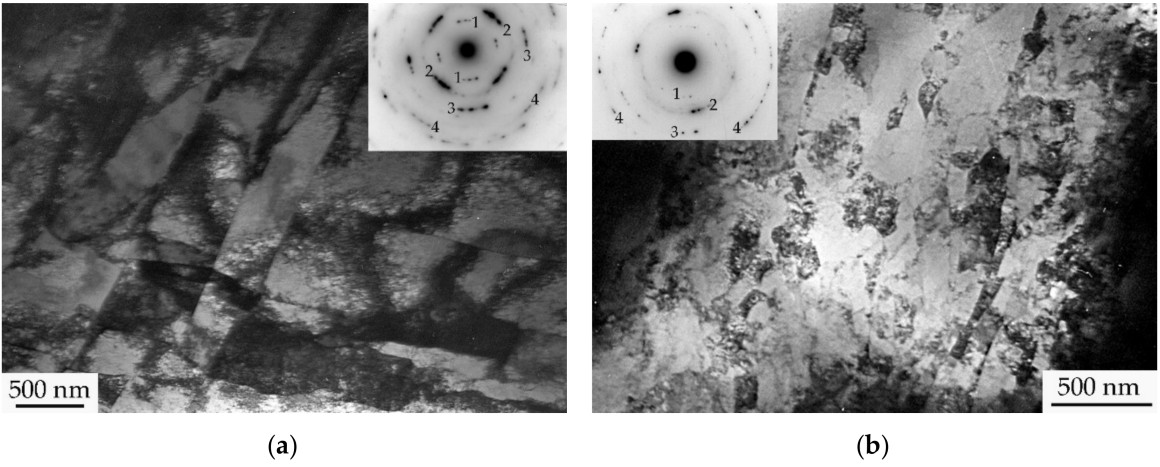

(**a**)　　　　　　　　　　　　　　　　　　　　　　　　　　　　(**b**)

**Figure 2.** Typical TEM bright-field images of microvolumes with most large-scale subgrains (**a**) and fine-grained structure (**b**) in $Ti_{50}Ni_{47.3}Fe_{2.7}$ after one ECAP pass at 723 K. Details are presented in text. Quasi-rings in microdiffraction patterns include the next types of B2-reflections: 1–(100), 2–(110), 3–(200), 4–(220), 5–(310) in (**a**); 1–(100), 2–(110), 3–(200), 4–(211) in (**b**).

The misorientation of adjacent band fragments reaches 10–12°, and that of adjacent fragments inside each band measures 2–5°. The band width in different microvolumes varies widely, from a microscale size of 400–500 nm (Figure 2a) to a mesoscale one of up to ~5 μm (Figure 2b). In mesoscale bands, one can clearly see a secondary microband structure with a minimum size of grain-subgrain fragments of 100–300 nm (Figure 2b).

The fragments of the grain-subgrain structure are mostly in the state of a B2 phase, as evidenced by electron diffraction patterns in Figure 2. It is observed that, along with quasi-ring reflections, the microdiffraction patterns reveal bright peaks from fragments misoriented to less than 15° with substantial radial broadening for microband volumes (Figure 2a), and with weak radial broadening and rather uniform distribution along the Debye rings for mesoband ones (Figure 2b). From comparison of the electron diffraction patterns in Figure 2a,b, it follows that the microstructure of the alloy after one ECAP pass is spatially inhomogeneous, not only in fragment sizes, but also in internal stresses. Moreover, the stress level in microband volumes (Figure 2a) is much higher than its level in mesoband ones (Figure 2b). Another feature of the electron diffraction patterns is a clear doublet structure of B2-phase reflections typical of deformation twinning. Besides the above microstructure types, individual grains of up to 1 μm occur, but rarely, in the alloy after one ECAP pass.

Therefore, the specimens of $Ti_{50}Ni_{47.3}Fe_{2.7}$ (at%) pressed in one ECAP pass at a channel angle of 90° assumes a grain-subgrain structure with a nonuniform fragment size distribution (100 nm to 1–1.5 μm) in which the main fraction belongs to submicrocrystalline fragments (100–500 nm). Such a grain-subgrain structure is formed through fragmentation on different scales and through B2-phase twinning.

Figures 3 and 4 shows the microstructure of the alloy after three ECAP passes. As can be seen, it is qualitatively similar to, but much finer than, the microstructure after one ECAP pass. The alloy preserves the system of microbands but their width decreases to less than 1 μm. The grain-subgrain structure contains a substantial fraction of nanosized fragments (50–100 nm). The main phase in the alloy is B2, as evidenced by its electron diffraction patterns.

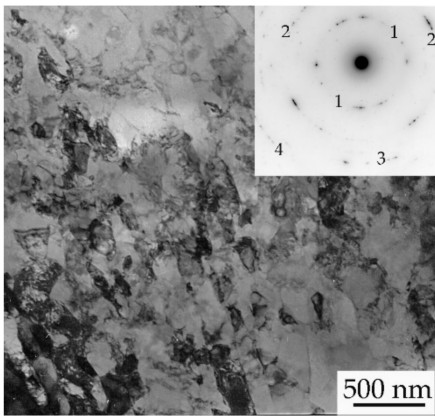

**Figure 3.** Typical microstructure in $Ti_{50}Ni_{47.3}Fe_{2.7}$ after three ECAP passes. The microdiffraction pattern is presented in the insert. Quasi-rings in microdiffraction pattern include the next types of B2 phase reflections: 1–(110), 2–(211), 3–(310), 4–(312).

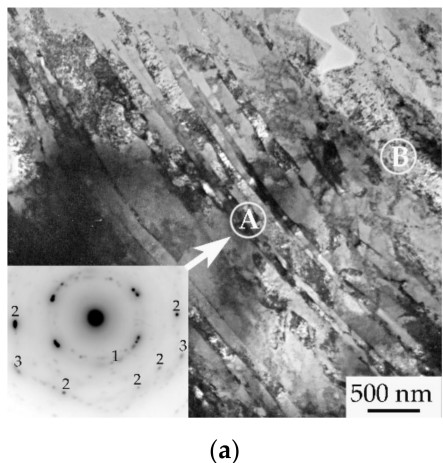
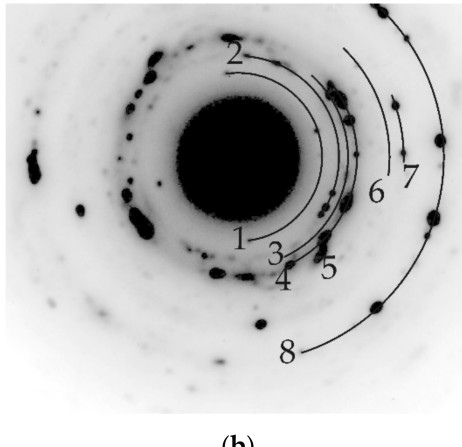

(**a**)   (**b**)

**Figure 4.** Microstructure in $Ti_{50}Ni_{47.3}Fe_{2.7}$ after three ECAP passes with the main B2 phase structure (region A) and small volume fraction of R phase (region B). The microdiffraction patterns are presented in the insert of (**a**,**b**), correspondingly. (**a**) The quasi-rings include the next types of B2 phase reflections: 1–(110), 2–(211), 3–(220); (**b**) The quasi-rings include the next types of B2 phase reflections: 1–(002), 2–(112), 3–(202), 4–(220), 5–(103), 6–(320), 7–(004), 8–(413).

However, several microvolumes with an R phase structure were found in the samples after three ECAP passes (e.g., region B in Figure 4a). It is most likely that the local appearance of the R phase at room temperature is due to the high level of residual stresses in these regions. According to electron microscopy data, the total proportion of regions with the R phase structure is less than 1 vol%. Therefore, the presence of these regions will not be taken into account in the subsequent discussion of the results.

Therefore, a mixed grain-subgrain structure based on a submicrocrystalline (100–500 nm) and a nanostructural fraction (50–100 nm) is formed in $Ti_{50}Ni_{47.3}Fe_{2.7}$ (at%) after ECAP at 723 K (channel angle 90°, strain rate 1 s$^{-1}$).

According to transmission electron microscopy [6], the microstructure of $Ti_{49.4}Ni_{50.6}$ (at%) after eight ECAP passes is also in the state of a B2 phase. The average grain-subgrain size in the specimen cross section is 300 nm.

## 3.3. Evolution of Structural Defects in TiNi-based Alloys Exposed to Warm ECAP

The evolution of structural defects was analyzed by positron annihilation spectroscopy. Figure 5 shows the average grain-subgrain size, average positron lifetime, and Vickers microhardness versus the numbers of ECAP passes for the test TiNi-based alloys.

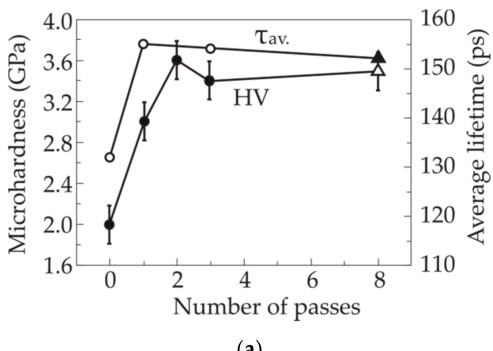
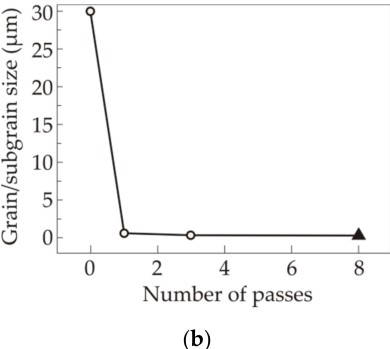

(**a**)   (**b**)

**Figure 5.** Average positron lifetimes, $\tau_{av.}$ (error bars ± 1 ps) and microhardness, HV, (**a**) and average grain/subgrain sizes (**b**) versus number of ECAP passes: $Ti_{49.4}Ni_{50.6}$ (at%) alloy (▲, △) and $Ti_{50}Ni_{47.3}Fe_{2.7}$ (at%) alloy (○, ●).

In the specimens exposed to ECAP, the average positron lifetime $\tau_{av}$ increases greatly even after the first ECAP pass and decreases slightly with increasing the number of passes irrespective of the alloy composition (Figure 5a). The Vickers microhardness, HV, behaves in the same way (Figure 5a). The increase in the microhardness can be explained by grain refinement in ECAP (Figure 5b), while the increase in $\tau_{av}$ definitely suggests that increasing the strain increases the lattice defect density. The type of defects resulting from ECAP can be identified by decomposing the positron lifetime spectra into components (Figure 6). The initial annealed specimens have a single-component spectrum with a positron lifetime $\tau_1 = 132 \pm 1$ ps for $Ti_{50}Ni_{47.3}Fe_{2.7}$ (at%) and $\tau_1 = 140 \pm 1$ ps for $Ti_{49.4}Ni_{50.6}$ (at%) (Figure 6a). The $\tau_1$ value corresponds to the experimental free positron lifetime 132 ps in TiNi [43] and the theoretical lifetimes of delocalized positrons in defect-free TiNi B2 structure: 120 ps [44] and 129 ps [45]. Such a decomposition shows that, even after the first ECAP pass, the spectra show no evidence of delocalized positrons: All positrons annihilate only from localized states associated with defects (so-called positron saturated trapping).

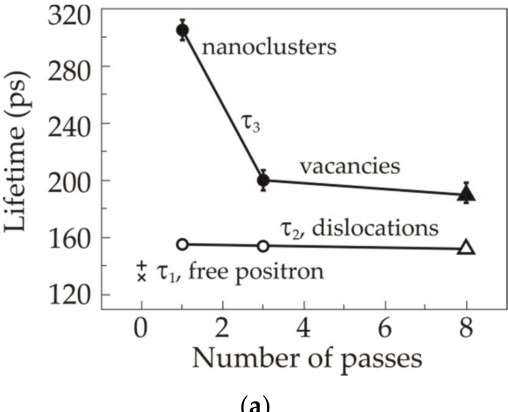 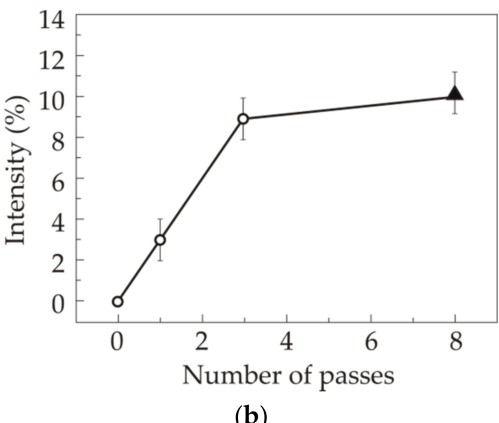

(a)  (b)

**Figure 6.** The positron lifetimes, corresponding to the different components of spectra; (**a**) and the intensities of $I_3$ component; (**b**) versus number of ECAP passes: $Ti_{49.4}Ni_{50.6}$ (at%) alloy (▲, △, +) and $Ti_{50}Ni_{47.3}Fe_{2.7}$ (at %) alloy (○, ●, ×).

Figure 6 demonstrates the evolution of positron lifetime components for $Ti_{50}Ni_{47.3}Fe_{2.7}$ (at%) and $Ti_{49.4}Ni_{50.6}$ (at%) alloys.

Noteworthy is that the positron lifetime component $\tau_2$ is invariant with the number of ECAP passes within experimental error (Figure 6a) and hence, the defect type identified with $\tau_2$ remains intact during ECAP strain accumulation. The same is observed in many experiments [46], and the $\tau_2$ values correspond to the lifetime of positrons trapping by dislocations.

The lifetime of $\tau_3$ component (Figure 6a) is far longer than the lifetime of delocalized positrons and the lifetime of positrons captured by dislocations. According to experimental data [47], the average positron lifetime in TiNi monovacancies is 195 ps. However, as has been shown by experiments [48,49] and theoretical studies [50], two types of vacancies differing in formation energy arise in B2 TiNi: vacancies $V_{Ti}$ and $V_{Ni}$ on its Ti and Ni sublattices, respectively. From first-principles calculations [50], it follows that the positron lifetimes in $V_{Ti}$ and $V_{Ni}$ vacancies differ greatly. The theoretical lifetime of positrons trapped by $V_{Ti}$ and $V_{Ni}$, vacancies equal 205 ps and 134 ps, correspondingly. This situation is distinct from what is observed in pure metals with only one vacancy type and in B2-structured transition metal aluminides with very close positron lifetimes in two types of sublattice vacancies [47]. The foregoing suggests that the component $\tau_3$ after three and eight ECAP passes corresponds to single $V_{Ti}$ vacancies. After the first ECAP pass, the positron lifetime $\tau_3 = 305$ ps (Figure 6a) being much longer than its value in monovacancies, can be related to nanoclusters of about five to seven vacancies, if judged from its theoretical dependence on the number of cluster vacancies in bcc metals [25]. Therefore, we have a nontrivial result since research data demonstrate that increasing the number of

ECAP passes increases both, the concentration and size of vacancy clusters in pure metals (copper, nickel, titanium) [30,51,52]. However, this conclusion follows from ECAP at room temperature when the diffusion mobility of vacancies is low and they likely fail to escape from the volume of grains to grain boundaries. The increase in vacancy concentration provides the conditions for vacancy clustering. Such an interpretation makes our results clear. After the first pass of warm ECAP, the grain-subgrain sizes remain rather large for vacancy clustering in the grain volume, but after the next passes, its level allows part of the vacancies to reach the boundaries of grains or dislocations, while the rest part falls short of the amount needed for the formation of noticeable concentrations of their complexes.

Figure 6b shows that the vacancy defect concentration associated with the intensity of the component $\tau_3$ increases steeply in three ECAP passes and then varies slightly. The intensities of the positron lifetime components $I_2$ and $I_3$ are proportional to the dislocation density and vacancy defect concentration ($I_2 + I_3 = 1$). As the saturated positron trapping, we cannot accurately estimate the dislocation density and vacancy defect concentration from the available positron trapping model [53]. However, we can estimate the vacancy concentration from the ratio $I_2/I_3 = v_d \rho_d / v_v c_v$, considering that the specific positron trapping rates by dislocations, and vacancies are equal, respectively to $v_d \sim 10^{-4}$ m$^2$s$^{-1}$ and $v_v \sim 10^{14}$ s$^{-1}$ in metallic materials [54]. The dislocation density for Ti$_{50}$Ni$_{50}$ (at%) after eight warm ECAP passes is $\rho_d = 1.1 \times 10^{15}$ m$^{-2}$ according to X-ray diffraction estimates [55]. The ratio $I_2/I_3$ equal to 10.1 and 9.1 after three, and eight ECAP passes, respectively, suggests that dislocations are the main type of defects for positron capture in TiNi-based alloys exposed to ECAP. At $I_2/I_3 \approx 10$, the estimated relative concentration of $V_{Ti}$ vacancies after ECAP is ~$10^{-4}$, being many orders of magnitude higher than their thermodynamic equilibrium concentration. Surely, the estimate is rough as the exact values of $v_d$ and $v_v$ are unknown for TiN-based alloys.

It stands to reason that $V_{Ni}$ vacancies are also produced in SPD processes, the more so their formation energy is 0.78 eV [48,49] against 0.97 eV for $V_{Ti}$ [48,49]. However, their concentration is unassessable by positron annihilation spectroscopy because of almost the same theoretical lifetimes of delocalized positrons and positrons in $V_{Ni}$ (134 ps). We expect that defect annealing experiments will help separate the contributions of dislocations and $V_{Ni}$ vacancies as they differ in annealing temperature.

## 4. Conclusions

Even one warm ECAP pass can provide submicrocrystalline fragmentation in initially coarse-grained TiNi-based alloys, and after three passes, their structure represents a mix of grains-subgrains sized to 100–500 nm and 50–100 nm. The dislocation defects and twins contribute to such refinement. The formation of UFG structures in Ti$_{50}$Ni$_{47.3}$Fe$_{2.7}$ (at%) does not change the martensite transformation sequence B2 $\leftrightarrow$ R $\leftrightarrow$ B19$'$ but it decreases the R $\leftrightarrow$ B19$'$ start temperature by 18 K and narrows the B19$' \rightarrow$ R temperature interval by 10 K. The B2 $\rightarrow$ R start temperature in this alloy after ECAP remains unchanged.

The refinement of the alloys to UFG structures under ECAP involves a substantial increase in the density of dislocations and vacancy-like defects. Their nanoclusters are detected only after the first ECAP pass, and only free volumes close in size to monovacancies are found after the next passes. The relative vacancy concentration estimated for the TiNi-based alloys after ECAP is about $10^{-4}$ being many orders of magnitude higher than their thermodynamic equilibrium concentration.

**Author Contributions:** Conceptualization, A.L. and A.B.; writing—original draft preparation, A.B.; writing—review and editing, A.L. and V.G.; software, R.L.; investigations, A.B., V.K., V.G. and R.L.; project administration, A.L.; funding acquisition, A.L. and R.L. All authors have read and agreed to the published version of the manuscript.

**Funding:** This research was funded by Fundamental Research Program of the State Academies of Science for 2013-2020 (project № III.23.2.2) and the grant of the Competitiveness Enhancement Program of National Research Tomsk Polytechnic University (VIU-OEF-177/2020).

**Conflicts of Interest:** The authors declare no conflict of interest.

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
