# Peer review of "Structural Defects in TiNi-Based Alloys after Warm ECAP"

_metals, doi:10.3390/met10091154_

Round 1

Reviewer 1 Report

  • Abstract should be rewritten and shortened.
  • There is no explanation about the importance of the ternary TiNiFe alloy in the Introduction. Basically, why was this alloy selected for the present research and compared with the binary alloy?
  • The philosophy behind the research has not been well expressed.
  • What was the ECAP rout?
  • I suggest that Results and discussion is started with “Effect of warm ECAP on martensite transformation temperatures in Ti50Ni47.3Fe2.7”, section 3.2.
  • Please index the main quasi-ring reflections or sharp spots in the SAED patterns.
  • Page 2, line 91: “The initial coarse-grained alloy had a grain size of 20–40 μm”. Is it grain size of primary austenite?
  • The main phase after ECAP processing was B2 (Page 4, lines 136 and 156). It means the microstructure contains R or B19’ phases. How do authors explain the occurrence of phase transformation during ECAP processing at 723 K.
  • Why did not the transformation temperatures and thermal hysteresis change after remarkable grain refinement and imposed strain after the first pass of ECAP processing? It is well-known that introducing defects during deformation leads to increase in thermal hysteresis.
  • Discussion on transformation temperatures after ECAP processing is poor and I believe that some related Refs should be added to support the data.
  • Using deg is not correct for K. Please remove deg in the text for example in Page 5 line 184 (you may write simply 18 K).
  • Data were not reported professionally in Fig. 4. Use the legend to show which plot belongs to which testing method. Error bars should be added to the data reported in Fig. 4.
  • Why did the microstructure and hardness (based on data shown in Fig. 4) reach a saturation only after the first pass of ECAP processing? I am wondering if dynamic recovery (or recrystallization) has key role in these results.
  • What is the relationship between the positron lifetimes results and microstructural observation revealed in Figs. 1 and 2 and phase transformation temperatures summarized in Table 1? Authors should provide a connection between these results in discussion.
  • I believe that it is NOT professional that the Results and discussion was terminated by referring to other works (Page 7, line 263).
  • The number of references is too high for such a paper 

Author Response

Point 1: Abstract should be rewritten and shortened.

Response 1: Abstract revised and shortened (page 1, lines 11-25)

Point 2: There is no explanation about the importance of the ternary TiNiFe alloy in the Introduction. Basically, why was this alloy selected for the present research and compared with the binary alloy?

Response 2: The authors gratefully accepted this comment and responded to it in the text of the manuscript (page 2, lines 77-82). The choice of this alloy for the study of the crystal structure defects after ECAP is due to the fact that at room temperature it is in the B2 phase state, since the start temperature of MT B2 → R TR = 275 K, and the start temperature of MT R → B19ʹ MS = 213 K. This circumstance allows us to study only those crystal structure defects which appear under the influence of ECAP, and to avoid the influence of defects that may appear as a result of MT. From the same considerations, a binary alloy Ti49.4Ni50.6 (at%) was selected, which also has MT temperatures below room temperature, and the sizes of the grains-subgrains after 8 passes of the ECAP wear the same as in the ternary alloy. The latter may be due to the fact that the ECAP of binary alloy was carried out on another installation with an angle between the channels of 110°.

Point 3: The philosophy behind the research has not been well expressed.

Response 3: We believe that the main focus of the work in the introduction has already been formulated, but it has received more clarity after answering the question about the rationale for choosing alloys for research.

Point 4: What was the ECAP rout?

Response 4: The ECAP route for both alloys was BC (page 1, line 15; page 2, line 87; page 3, line 98).

Point 5: I suggest that Results and discussion is started with “Effect of warm ECAP on martensite transformation temperatures in Ti50Ni47.3Fe2.7”, section 3.2.

Response 5: We agree with the comment. The discussion of the research results is started in accordance with the reviewer's recommendation: with the influence of the warm ECAP on the MT temperatures in The Ti50Ni47.3Fe2.7 (at%) alloy.

Point 6: Please index the main quasi-ring reflections or sharp spots in the SAED patterns.

Response 6: We agree with the comment. The microdiffraction patterns (figures 2, 3 and 4) are indexed (page 4, lines 153-155; page 5, lines 178-187).

Point 7: Page 2, line 91: “The initial coarse-grained alloy had a grain size of 20–40 μm”. Is it grain size of primary austenite?

Response 7: Yes. This is the grain size in the high-temperature B2 phase. The corresponding clarification was made in the text of the manuscript. (page 2, line 89)

Point 8: The main phase after ECAP processing was B2 (Page 4, lines 136 and 156). It means the microstructure contains R or B19ʹ phases. How do authors explain the occurrence of phase transformation during ECAP processing at 723 K.

Response 8: The authors are grateful to the reviewer for this question. Indeed, after the ECAP in some places of the sample, reflections are observed that correspond to the R phase reflexes. This phase may appear at cooling after ECAP under the influence of internal stress concentrators. These stress concentrators are inevitably formed in the samples after the ECAP, despite relaxation processes at the ECAP temperature of 723 K. Corresponding changes were made to the text of the manuscript. (Figure 4 and page 6, lines 193-198)

Point 9: Why did not the transformation temperatures and thermal hysteresis change after remarkable grain refinement and imposed strain after the first pass of ECAP processing? It is well-known that introducing defects during deformation leads to increase in thermal hysteresis.

Response 9: This is impotent question. However, the answer is not simple. A similar result on the effect of intense plastic deformation by abc pressing at the same temperature of 723 K on the temperatures and sequence of MT in Ti49.8Ni50.2 (at%) alloy samples was observed earlier. Even with the true strain value e = 8.4, the forward and reverse MT temperatures within the measurement error did not change. Therefore, we are confident that the results presented in our manuscript are correct. We assume that these results are due to the following. The forging temperature at abc and ECAP is sufficient to ensure diffusive mobility of all types of defects in the crystal structure. Therefore, we think that the observed results are due to relaxation processes at the deformation temperature. However, we agree with the reviewer that the introduction of defects in the samples should lead to hardening and hindering the movement of the martensitic phase domains. Therefore, we currently do not have a clear and unambiguous answer to the reviewer's question.

Point 10: Discussion on transformation temperatures after ECAP processing is poor and I believe that some related Refs should be added to support the data.

Response 10: The main purpose of the work is to study the crystal structure defects that are formed after warm ECAP in TiNi-based alloys. It is noted in paper the influence of the ECAP on the MT temperatures in the studied alloys and the fact that the temperature of the beginning of the MT in these alloys are below room temperature. The latter allows exclude the influence of defects in the crystal structure, which they may appear with MT, based on the results of our research. Therefore, we ask the reviewer to allow us leave the text of the manuscript in this part unchanged.

Point 11: Using deg is not correct for K. Please remove deg in the text for example in Page 5 line 184 (you may write simply 18 K).

Response 11: We agree with the comment. The corresponding changes were made in the manuscript. (page 1, line 44; page 4, lines 132, 137, 138; page 8, line 283)

Point 12: Data were not reported professionally in Fig. 4. Use the legend to show which plot belongs to which testing method. Error bars should be added to the data reported in Fig. 4.

Response 12: We agree with the comment. Corresponding changes have been made to the figure caption and to figure 4. (Figure 5 in revised text; page 6, line 209)

Point 13: Why did the microstructure and hardness (based on data shown in Fig. 4) reach a saturation only after the first pass of ECAP processing? I am wondering if dynamic recovery (or recrystallization) has key role in these results.

Response 13: The authors of the manuscript, as well as the reviewer, believe that the results obtained are due to the processes of dynamic recrystallization.

Point 14: What is the relationship between the positron lifetimes results and microstructural observation revealed in Figs. 1 and 2 and phase transformation temperatures summarized in Table 1? Authors should provide a connection between these results in discussion.

Response 14: In our opinion, the relationship between changes in various components of the positron lifetime with the evolution of the microstructure as a result of ECAP is sufficiently reflected in the text of the work. First of all, it is noted that in the initial state of coarse-grained samples, there is only one component of the positron lifetime τ1, which corresponds to the annihilation of positrons in the defect-free B2 phase (Page 6). Nanoclusters are formed only after the first ECAP pass. On page 7 (lines 253-256) it is indicated that with an increase in the number of passes (and, accordingly, a decrease in the size of grains-subgrains), the formation of nanoclusters does not occur. Some vacancies reach the grain-subgrain boundaries and dislocations. In this case, only the components of the positron lifetime τ2 and τ3 are observed, which are caused by the annihilation of positrons trapped by vacancies in the Ti B2 phase sublattice and by dislocations.

Point 15: I believe that it is NOT professional that the Results and discussion was terminated by referring to other works (Page 7, line 263).

Response 15: We think that this form of the reviewer's statement is too harsh. However, the authors agree with the comment on the substance of the issue and made changes to the manuscript: this sentence is removed.

Point 16: The number of references is too high for such a paper.

Response 16: The authors have tried to reflect the contributions of other researchers to the study of the issues discussed in this manuscript, and ask the reviewer to soften their attitude to the number of references to the work of other authors.

Reviewer 2 Report

A Lotkov et al.

This paper investigates the effects of ECAP on microstructures, martensitic transformations and lattice defects evolution in Ti-Ni-Fe and T-Ni alloys. The obtained results seem to be probable, but the following points should be considered for further improvement of this paper.

  1. No explanation why the ternary and binary alloys can be directly compared in Figures 4 and 5; nevertheless different numbers of ECAP cycles and channel angle of ECAP die. If that is the case, furthermore, how the temperatures of Ms, As and Af change in Ti-Ni alloy after eight ECAP passes? Are those continuously connecting to the temperatures in Table 1? The effects of iron addition to the binary system not only lattice defects evolution but also martensitic transformations will be also reader’s concern.
  2. In this paper, no consideration about the possibility of positron annihilation captured by grains/subgrains in Figures 1 and 2 has been made. What is the lifetime of the positron?

Author Response

Point 1: No explanation why the ternary and binary alloys can be directly compared in Figures 4 and 5; nevertheless different numbers of ECAP cycles and channel angle of ECAP die. If that is the case, furthermore, how the temperatures of MS, AS and Af change in Ti-Ni alloy after eight ECAP passes? Are those continuously connecting to the temperatures in Table 1? The effects of iron addition to the binary system not only lattice defects evolution but also martensitic transformations will be also reader’s concern.

 Response 1: The choice of Ti50Ni47.3Fe2.7 (at%) alloy for the study of crystal structure defects after ECAP (the angle between channels is 90°) is due to the fact that at room temperature it is in the B2 phase state, since the start temperature of MT B2 → R TR = 275 K, and the start temperature of MT R → B19′ MS = 213 K. This circumstance allows us to study only those defects of the crystal structure which appear under the influence of ECAP, and to avoid the influence of defects that may appear as a result of MT. From the same considerations, the binary alloy Ti49.4Ni50.6 (at%) was selected, which also has MT temperatures below room temperature, and the size of the grains-subgrains after 8 passes of the ECAP was the same as in the ternary alloy. The latter is due to the fact that ECAP of binary alloy was performed on a different installation with an angle between the channels of 110° and to achieve a microstructure similar to that in the ternary alloy, more ECAP passes were required. In both cases, the ECAP route was BC. The high degree of similarity of microstructures in binary and ternary alloys after ECAP at the same pressing temperature gives us the opportunity to study the crystal structure defects in them as in a single system.

The result of the effect of severe plastic deformation on the MT temperatures depends on the deformation temperature of the samples. In our case, the temperature of the ECAP for both alloys is equal to 723 K, which is sufficient to ensure the diffusion mobility of all types of defects in the crystal structure. Our results show that in this case, the MT temperatures do not change as the true strain given to the samples increases up to e = 2.7. A similar result on the effect of severe plastic deformation by abc pressing at the same temperature of 723 K on the temperatures and sequence of MT in Ti49.8Ni50.2 (at%) alloy samples was observed earlier. Even with the true strain value e = 8.4, the forward and reverse MT temperatures within the measurement error did not change. We assume that the observed results are due to relaxation processes at a given deformation temperature.

Point 2: In this paper, no consideration about the possibility of positron annihilation captured by grains/subgrains in Figures 1 and 2 has been made. What is the lifetime of the positron?

Response 2: According to [Staab, T. E. M., Krause-Rehberg, R. & Kieback, B. Review Positron annihilation in fine-grained materials and fine powders—an application to the sintering of metal powders. Journal of Materials Science 34, 3833–3851 (1999). https://doi.org/10.1023/A:1004666003732 ], dislocations and small-angle boundaries have very close lifetimes, and are usually not separated. The lifetime of positrons, τHAB, in the vicinity of large-angle boundaries is τHAB = 2.7 τbulk [Staab, T. E. M., Krause-Rehberg, R. & Kieback, B. Review Positron annihilation in fine-grained materials and fine powders—an application to the sintering of metal powders. Journal of Materials Science 34, 3833–3851 (1999). https://doi.org/10.1023/A:1004666003732 ]), where τbulk = τ1 (the lifetime of the delocalized positrons in defect-free B2 phase in the present manuscript). The τ1 lifetime are 132 ps and 140 ps in Ti50Ni47.3Fe2.7 and Ti49.4Ni50.6 (at%) alloys, correspondingly. Hence, τHAB are about 356 ps and 378 ps in ternary and binary our alloys. We do not observe such components in our positron lifetime spectra.

Round 2

Reviewer 1 Report

I have no further comments on the paper. I believe it should be accepted in the present form.

Reviewer 2 Report

None